# SpareTrain: Fault-Tolerant LLM Training via Low-Cost Dual Modular Redundancy

**Rihae Park, Yeonjae Kim, Seung Yul Lee, Yeonhong Park, Jae W. Lee**
Seoul National University
{rihae,duswo1120,triomphant1,ilil96,jaewlee}@snu.ac.kr

## Abstract

Dual Modular Redundancy (DMR) is a highly effective mechanism for detecting silent data corruption (SDC)—a critical reliability concern in large language model (LLM) training—by executing each operation twice. However, its high computation overhead has prevented practical deployment at scale. In this paper, we present SpareTrain, an LLM training system that achieves complete DMR with minimal overhead by repurposing the activation checkpointing mechanism and exploiting idle GPU time. Evaluations on up to 32 H200 GPUs show that SpareTrain improves throughput by 12–35% over naïve DMR, corresponding to only 3–14% overhead compared to unprotected training, while maintaining full DMR error detection capabilities.

## 1 Introduction

Large language models (LLMs) are transforming daily life (Grattafiori et al., 2024; Achiam et al., 2023; Yang et al., 2024). Behind this remarkable success lies the tremendous training cost. Training LLMs requires massive compute clusters and often takes several months to complete over large-scale datasets (Grattafiori et al., 2024; Achiam et al., 2023; Yang et al., 2024; Isaev et al., 2023; Laurençon et al., 2022).

Recently, reliability has become a serious concern for LLM training, as even a single bit-flip can derail months-long runs. Particularly, addressing Silent Data Corruption (SDC)—errors not caught by intrinsic hardware mechanisms such as ECC or CRC—has emerged as a critical challenge due to the difficulty of detecting them (He et al., 2023; Ma et al., 2025; Bonderson, 2021).

Dual Modular Redundancy (DMR), which executes the same computations twice and compares the outputs, is the most effective way to detect SDCs (Reinhardt & Mukherjee, 2000; Jeon & Annavaram, 2012; Dixit et al., 2021; Ma et al., 2025). Despite its completeness, the use of DMR in many practical settings is limited due to its overhead. DMR effectively doubles the computational cost. This leads to the key research question that this paper addresses: how can we minimize the LLM training throughput loss caused by DMR without compromising its SDC detection capability?

To answer this research question, this paper investigates two key strategies: Piggyback-DMR (P-DMR) and Deferred-DMR (D-DMR). P-DMR piggybacks DMR onto the inherent redundancy introduced by activation checkpointing—a de facto standard for memory savings in LLM training (Isaev et al., 2023; Liang et al., 2025a; MosaicML, 2023; Narayanan et al., 2021). D-DMR leverages the fact that GPU compute idle cycles constitute a significant portion of training time, primarily due to the overheads of parallelism strategies used in distributed training. By moving redundant computations for DMR into these idle cycles, it becomes possible to minimize the latency increase due to DMR.

Building on these two strategies, we propose SpareTrain, which systematically combines P-DMR and D-DMR to maximize latency savings while carefully managing their potential side effect: memory overhead. In our evaluation on Llama-3-70B, Mistral-Large, and Llama-4-Scout (70B–123B parameters) using up to 32 H200 GPUs under various memory setups, SpareTrain improves the throughput of a DMR-protected LLM training system by up to 35%, 29%, and 16%, respectively, while fully preserving detection capability. These gains translate to only 3–14% overhead compared to baseline training without SDC protection.

## 2 BACKGROUND

### 2.1 SILENT DATA CORRUPTIONS IN LLM TRAINING

Silent Data Corruptions (SDCs) are hardware-induced errors that evade detection, allowing corrupted values to propagate silently and compromise application correctness. These errors have become increasingly problematic due to factors such as the growing fragility of hardware components from aggressive technology scaling. Processors such as CPUs and GPUs are more vulnerable to SDCs than memory systems because they have limited protection mechanisms, making them the primary source of SDCs in modern computing systems (He et al., 2023; Mitra et al., 2025; Dixit et al., 2021; 2022).

Although SDCs are rare, they pose a significant risk to LLM training in datacenters, where runs can span months across hundreds of thousands of GPUs (Bonderson, 2021). Recent technical reports and industry papers indicate that when they occur, SDCs can significantly degrade model accuracy and destabilize training convergence (He et al., 2023; Laurençon et al., 2022; Grattafiori et al., 2024; Team et al., 2023). While SDC-induced gradient noise within a single optimizer step may appear negligible, large-scale studies show that its accumulation over time can cause parameter divergence and convergence to yield suboptimal models (Ma et al., 2025). Other studies further challenge the conventional belief that small-magnitude errors are harmless, showing that even minor SDCs can irreversibly degrade model quality (He et al., 2023).

### 2.2 SOFTWARE TECHNIQUES FOR DETECTING SDCS

Software-level SDC detection spans a spectrum of approaches, trading detection guarantees against computational efficiency. At one end are expensive *exact* techniques; at the other are *lightweight* methods that lower overhead at the cost of incomplete coverage.

**Dual Modular Redundancy**. Dual Modular Redundancy (DMR) detects errors by executing each operation twice—once as the *primary execution* and once as the *checker execution*—and comparing their results. A key advantage of DMR is its generality—it can be applied to any type of operation, whether linear or non-linear. Under the standard assumption that identical faults in both executions are negligibly probable, DMR achieves complete error detection. However, DMR also has a critical limitation: it incurs substantial computational overhead—approximately 100%—since each operation must be performed twice.

For decades, efficient DMR implementations have been extensively studied across hardware, compiler, and software layers (Austin, 1999; Reinhardt & Mukherjee, 2000; Mukherjee et al., 2002; Yim et al., 2011; Wang et al., 2007; Wadden et al., 2014; Reis et al., 2005; Didehban & Shrivastava, 2016; Oh et al., 2002; Jeon & Annavaram, 2012; Abdel-Majeed et al., 2015). However, none of these works target large-scale LLM training, despite its growing importance for reliability. Industry reports have mentioned the use of DMR-like techniques for enhancing reliability, underscoring its potential relevance in this domain (Team et al., 2023; Ma et al., 2025). Nevertheless, likely due to the high overheads, no deployment has achieved full coverage of DMR.

**Algorithm-Based Fault Tolerance**. For linear algebra kernels such as General Matrix Multiplication (GEMM), Algorithm-Based Fault Tolerance (ABFT) (Huang & Abraham, 1984) offers a more efficient alternative by embedding checksum-based invariants into the computation. ABFT can detect and, in some cases, correct errors with far lower overhead than DMR, assuming at most one fault occurs per GEMM invocation. However, when applied in conjunction with reduced-precision formats such as FP16, BF16, and FP8—which have become the de facto standard technique in LLM training—the limited numerical precision can compromise the accuracy of ABFT, potentially leading to false negatives (Ma et al., 2025). As a result, despite its high computational overhead, DMR remains the more robust and reliable option for building SDC-free LLM training systems.

**Approximate Approaches**. A number of recent works propose even lighter-weight, non-exact methods tailored to ML workloads. For example, He et al. (2023) flags extreme gradient outliers as potential SDCs, and other approaches train small neural networks as SDC detectors (Ma et al., 2024). A more specialized line of work focuses on LLMs, such as Liang et al. (2025b), which detects and corrects anomalous outputs with just 7% end-to-end overhead by targeting only computation in specific layers like attention. While these methods demonstrate promising results, their selective

coverage and probabilistic nature limit their applicability to production environments that require strong reliability guarantees.

Among software-level techniques, DMR remains the only method that can guarantee complete error detection for arbitrary LLM training workloads. However, its perceived high computational cost has led to systematic exclusion from practical large-scale training considerations. This motivates our work, which aims to make DMR practical without sacrificing its detection guarantees.

## 3 STRATEGIES FOR REDUCING DMR OVERHEAD IN LLM TRAINING

A naïve application of DMR would roughly double each operation's execution time due to checker execution overhead. However, large-scale LLM training exhibits characteristics that can mitigate this cost. In this section, we identify two complementary strategies: 1) *Piggyback-DMR (P-DMR)*, which exploits redundancy already present in activation checkpointing, and 2) *Deferred-DMR (D-DMR)*, which utilizes GPU idle periods to hide checker execution overheads.

### 3.1 P-DMR: LEVERAGING REDUNDANCY IN ACTIVATION CHECKPOINTING

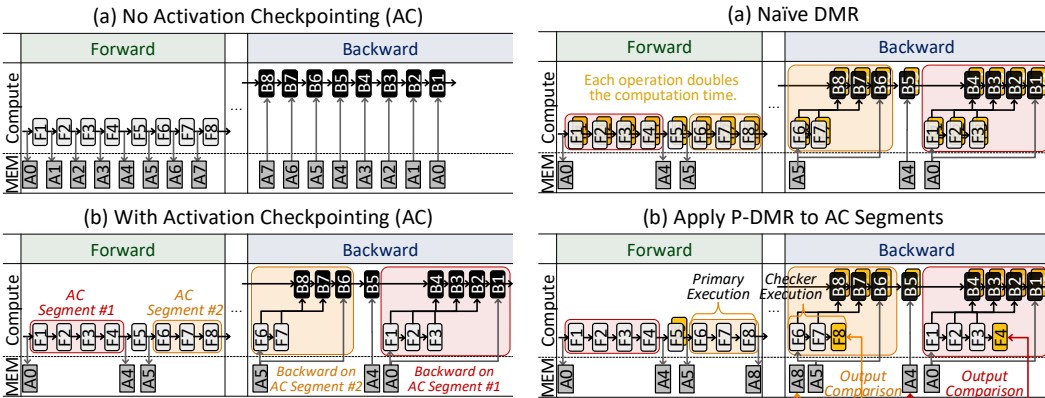

Figure 1: Activation Checkpointing (AC)          Figure 2: Naïve DMR vs. Piggyback-DMR

**Opportunity: Activation Checkpointing**. Activation Checkpointing (AC) is a popular technique for reducing memory overhead from intermediate activations. Figure 1 illustrates its concept by comparing execution without AC (a) and with AC (b). With AC, consecutive forward operations are grouped into AC segments (e.g., F1–F4, F6–F8), where only their inputs—known as checkpoints (e.g., A0, A5)—are stored, while the other intermediate activations are discarded and recomputed during the backward pass. AC can be applied fully, where all operations form a single segment, or selectively, where multiple small segments are created and some operations (e.g., B5) remain outside any segment. How segments are formed depends on factors such as the memory budget and model structure, but AC has become a de facto standard in LLM training for its substantial memory savings (Isaev et al., 2023; Liang et al., 2025a; MosaicML, 2023; Narayanan et al., 2021).

**Proposed Strategy: Piggyback-DMR (P-DMR)**. Under AC, all operations within a segment are re-executed during the backward pass, except for the last operation whose outputs are typically already retained. Piggyback-DMR (P-DMR) leverages these recomputations as natural checker executions: the forward pass serves as the primary execution, while the backward recomputation—augmented with one extra run of the last operation—serves as the checker execution. The resulting outputs are then compared with those from the forward pass, completing DMR coverage for the entire segment. Figure 2 contrasts naïve DMR and P-DMR under the same AC segment configuration as Figure 1, with additional computation for DMR shown in yellow. In naïve DMR, every operation requires its own checker execution, doubling the computation time. In contrast, P-DMR adds only one extra execution of the last operation per AC segment—F8 or F4 in the backward pass—completing the checker execution for the segment. Since the outputs of each segment's last operation are already stored in the forward pass in most cases, retaining them for comparison incurs no additional memory overhead. Detailed discussion is provided in Appendix A.

## 3.2 D-DMR: LEVERAGING GPU IDLE TIME

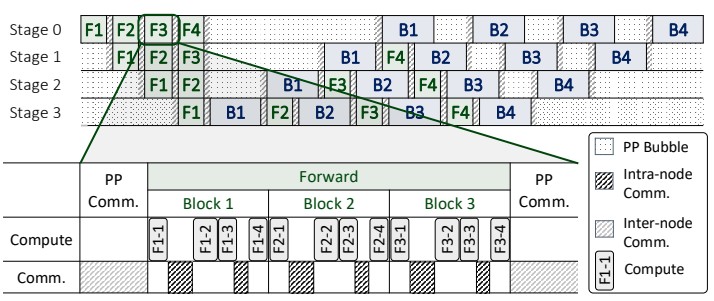

Figure 3: Compute and communication timeline of LLM training.

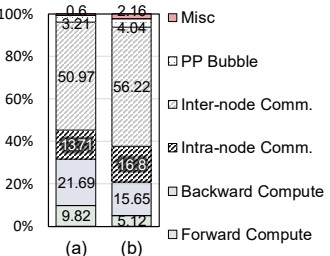

Figure 4: Training time breakdown for (a) Mistral-Large and (b) Llama-4-Scout

**Opportunity: GPU idle time**. Training LLMs requires large-scale GPU clusters that employ multiple parallelism strategies, including data parallelism (DP), pipeline parallelism (PP), tensor parallelism (TP), and expert parallelism (EP) for Mixture of Experts (MoE) models. While essential for scaling, these strategies introduce substantial GPU compute idle time from two main sources (Isaev et al., 2023; Liu et al., 2023b; Pati et al., 2023; Wang et al., 2024; Feng et al., 2025). The first is communication overhead, both intra- and inter-node. TP and EP involve frequent, high-volume communication, so they are typically confined within a single node, relying on high-bandwidth intra-node interconnects (Jin et al., 2025; Liu et al., 2025). DP and PP often span across nodes. Second, pipeline stage dependency in PP forces GPUs to wait for preceding stages, creating idle periods known as PP bubbles. Such communication patterns and PP bubbles are illustrated with an example of LLM training with four PP stages in Figure 3.

We quantify idle time by analyzing the training time breakdown of two representative LLM architectures: Mistral-Large (a dense model) and Llama-4-Scout (an MoE model), as shown in Figure 4. Both models are trained on clusters of H200 nodes (8 GPUs per node). Mistral-Large is configured with PP=3, TP=8, and batch size=64 on three nodes, while Llama-4-Scout is configured with PP=4, TP=8 for non-expert layers, EP=8 for experts, and batch size=64 on four nodes. Both models exhibit substantial idle time (68% and 77%, respectively), which corresponds to the combined fraction of communication and PP bubbles.

**Proposed Strategy: Deferred-DMR (D-DMR)**. Deferred-DMR (D-DMR) leverages GPU idle time to hide checker execution overhead. Instead of running checker executions immediately after primary executions (as in naïve DMR), D-DMR defers them to later idle periods. While this reduces throughput loss, it introduces memory overhead since input and output pairs must be retained until checker executions, requiring careful balance between performance and memory.

In D-DMR, only communication-induced idle time is exploited, as PP bubbles are difficult to utilize due to their infrequency and uneven distribution (see Figure 3). Thus, D-DMR defers checker executions by overlapping them with communication windows, and further distinguishes between D-DMR$_{inter}$, which overlaps with inter-node communication (e.g., PP), and D-DMR$_{intra}$, which overlaps with intra-node communication (e.g., TP, EP).

## 4 SPARETRAIN OVERVIEW

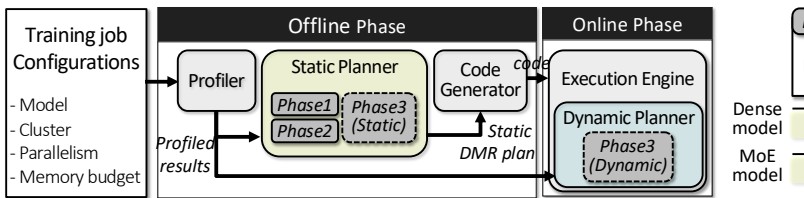

Figure 5: SpareTrain overview.

Figure 6: Three phases of DMR planning.

We present SpareTrain, a novel framework for lightweight, SDC-resilient LLM training that enables full DMR with minimal overhead. The central design question for SpareTrain is determining how to utilize P-DMR and D-DMR to minimize DMR cost, which we refer to as a *DMR plan*. A DMR plan assigns each operation to one of four sets: *P-DMR*, *D-DMR$_{inter}$*, *D-DMR$_{intra}$*, or *Naïve-DMR*, corresponding to the sets of operations verified by P-DMR, D-DMR$_{inter}$, D-DMR$_{intra}$, and naïve DMR, respectively. For operations assigned to *D-DMR$_{inter}$* and *D-DMR$_{intra}$*, the planner must also determine the specific idle window for placing the checker execution.

Figure 5 presents a high-level overview of SpareTrain. Notably, SpareTrain consists of both an offline and an online component, each with its own planner: a static planner and a dynamic planner, both responsible for generating DMR plans.

In the offline phase, a profiler executes a few iterations of the training job under the given configuration to collect information required for planning. This includes (1) the activation checkpointing (AC) configuration, i.e., how operations have been segmented into AC segments, (2) the execution time and memory usage of each operation, (3) the timing and duration of GPU idle windows, and (4) per-device memory usage patterns. The static planner then uses this information to generate a plan and rewrite the training job code accordingly.

During runtime, the execution engine runs the generated code. If a mismatch is detected on any device during DMR checks, the entire iteration is rolled back and re-executed. If there are operations that cannot be planned offline (e.g., MoE layers), the dynamic planner monitors computation and communication at runtime to dynamically determine and execute DMR plans, opportunistically applying D-DMR.

## 5 DMR PLANNING

Figure 6 shows a high-level overview of the DMR planning process in SpareTrain, which forms its core mechanism. DMR planning in SpareTrain is composed of three phases. In the first phase, P-DMR is planned by assigning operations to *P-DMR*. The second phase performs coarse-grained assignment for D-DMR, grouping operations and assigning them to *D-DMR$_{inter}$*. The third phase then handles the remaining operations with fine-grained D-DMR, assigning each operation individually to either *D-DMR$_{inter}$* or *D-DMR$_{intra}$*. Any operations not assigned in these phases naturally fall back to *Naïve-DMR*. For non-MoE models, where all operations are fixed, all three phases are performed by the static planner. For MoE models, however, Phase 3 is carried out by the dynamic planner to handle iteration-dependent variability.

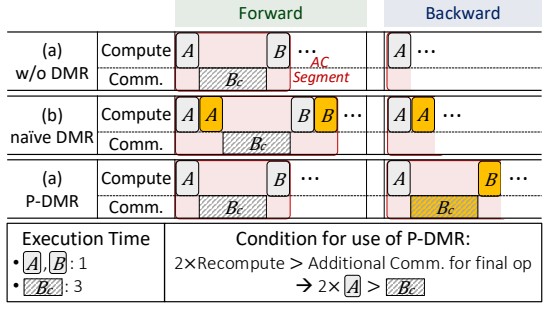

Figure 7: Exception case of AC segment where naïve DMR outperforms P-DMR.

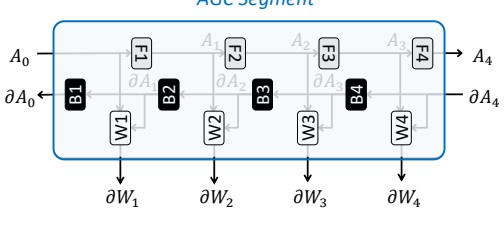

$$A_0 - A_4, \partial A_0 - \partial A_4, \partial W_4 - \partial W_4 \quad (14)$$
$$A_0, A_4, \partial A_0, \partial A_4, \partial W_4 - \partial W_4 \quad (8)$$

w/o Checkpointing: $A_0 - A_4, \partial A_0 - \partial A_4, \partial W_4 - \partial W_4$ (14)
w/ Checkpointing: $A_0, A_4, \partial A_0, \partial A_4, \partial W_4 - \partial W_4$ (8)

Figure 8: Concept of Activation Gradient Checkpointing (AGC).

### 5.1 PHASE 1: PLANNING FOR P-DMR BY STATIC PLANNER

**General Rule**. By default, the static planner applies P-DMR for all operations recomputed by AC. P-DMR is usually beneficial since the only extra cost per AC segment is the last operation (e.g., F8, F4 in Figure 2(b)).

**Exception Case**. In certain cases, naïve DMR is preferable to P-DMR. This occurs when the last operation of an AC segment requires expensive communication to gather its input operands. With

naïve DMR, since the two redundant executions occur back-to-back, the input operands need to be gathered only once and can then be reused. In contrast, P-DMR requires replaying the communication because the two executions are separated in time.

Figure 7 illustrates such an exception case, comparing execution without DMR (a), naïve DMR (b), and P-DMR (c) for an AC segment including two operations $A$ and $B$, where $B$ involves communication denoted as $B_c$. The additional overhead of naïve DMR is $2 \times A + B$. P-DMR eliminates the duplication of $A$ but requires replaying $B_c$, resulting in $B_c + B$. When $B_c > 2 \times A$, naïve DMR is preferable, so the planner selects it. While this example segment is small with only operation A being recomputed, in general, A can represent the collection of all operations that are recomputed.

**For MoE Models**. For AC segments consisting only of static operations, the same procedure applies as in non-MoE models. If dynamic operations (e.g., MoE layers) are included in an AC segment, exact cost calculation and comparison may not be feasible, so the planner defaults to *P-DMR*.

## 5.2   PHASE 2: COARSE-GRAINED PLANNING FOR D-DMR BY STATIC PLANNER

**Activation-Gradient Checkpointing (AGC)**. To make D-DMR efficient, the planner first applies a coarse-grained strategy: it groups multiple connected operations in the computation graph into a single segment and defers them together. In this way, only the segment's boundary tensors (inputs/outputs) are needed to replay the entire segment and verify results. We call this *Activation–Gradient Checkpointing (AGC)*, by analogy to activation checkpointing (AC), but extended to the backward pass, where both activations and gradients are checkpointed.

Figure 8 illustrates the idea on a simple graph with four forward ops (F*) and their corresponding backward (B*) and weight-gradient (W*) operations. Under operation-level deferral (every operation deferred individually), the inputs and outputs of *each* operation must be retained—14 tensors in total. In contrast, treating the same set as one AGC segment requires retaining only the boundary tensors—8 in total. During the checker execution, the segment is replayed from the stored inputs, and the computed outputs are compared with the stored outputs, completing DMR.

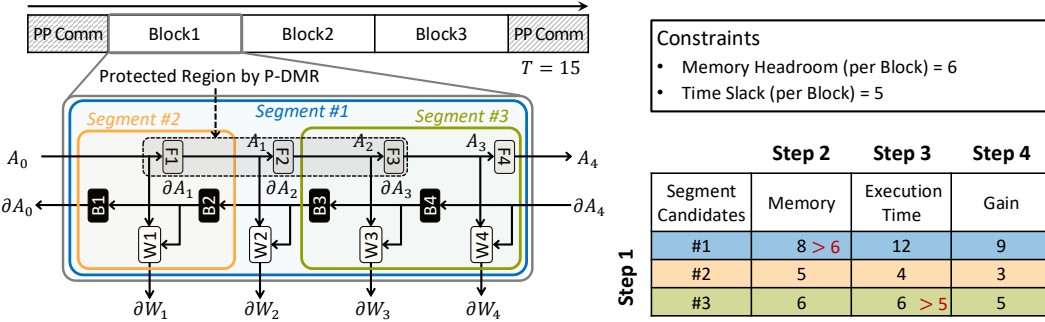

Figure 9: AGC segment selection process.

**AGC-Based D-DMR Planning**. AGC greatly reduces D-DMR's memory overhead but introduces additional complexity in configuring AGC segments, and D-DMR$_{intra}$ windows are typically too short to accommodate sizeable segments. Therefore, the static planner selects only *one* AGC segment per transformer block and assigns it to *D-DMR$_{inter}$*, scheduling it in the PP communication window immediately after the corresponding backward stage.

Accordingly, the goal of this phase is to select the AGC segment that maximizes time saving for checker executions, subject to two constraints: (i) *memory headroom*, defined as the available budget beyond the peak usage from the stage through the following PP communication, and (ii) *time slack*, the inter-node communication window available to the block. While a larger segment offers greater savings, it also increases the likelihood of violating memory or time slack constraints, necessitating a careful selection.

The selection process consists of four steps: Step 1 enumerates all possible segments within a block; Step 2 filters out segments that exceed the memory headroom constraint; Step 3 eliminates those that violate the time slack constraint; and Step 4 selects, from the remaining candidates, the segment

with the greatest gain. Here, gain is defined as the *checker execution time saved*: the replay time of the segment minus any extra costs (e.g., operations already protected by P-DMR, or additional intra-segment communication).

Figure 9 illustrates the overall selection process. Unit memory and execution time are assumed for all tensors and operations. The stage has a total memory headroom of 18 and an upcoming PP communication time of 15. Dividing evenly across three blocks gives a memory headroom of $18 \div 3 = 6$ and a time slack of $15 \div 3 = 5$ per block. Three candidate segments are shown for brevity. Segment #1 violates the memory constraint (requires 8 tensors: $A_0, A_4, \partial A_0, \partial A_4, \partial W_1 - \partial W_4$). Segment #3 violates the time slack constraint (replay time $= 6$: F3, B3, W3, F4, B4, W4). Segment #2 is feasible and thus compared against other feasible candidates; among them, the one yielding the largest gain is selected. For Segment #2, the replay time is 4 with an extra cost of 1 from re-protecting F1, yielding a gain of 3 ($= 4 - 1$). An example selection result from our evaluation is provided in Appendix B.

**For MoE Models**. AGC segments are selected only from the static parts of the computation graph, and memory headroom is conservatively set based on the maximum usage observed during profiling.

### 5.3 PHASE 3: FINE-GRAINED PLANNING FOR D-DMR BY STATIC OR DYNAMIC PLANNER

Phase 3 assigns the operations not covered by Phases 1–2 in a fine-grained manner (i.e., per-operation). These remaining operations are considered for deferral into unused communication windows, including intra-node communication time within each stage or inter-node (PP) communication time immediately following each stage that remains after Phase 2. In this phase, operation-level D-DMR is managed through a D-DMR queue. How operations are enqueued and scheduled differs between non-MoE and MoE models.

**For Non-MoE Models**. For non-MoE models, the static planner executes this phase. It scans operations in order of execution. When a not-yet-used communication window (scheduling point) is reached, the following steps are performed: Step 1 tentatively defers all unassigned operations since the last scheduling point and checks whether the resulting D-DMR queue exceeds the memory budget. Step 2 keeps all operations deferred if within budget; otherwise, demotes shortest-runtime operations to *Naïve-DMR* until the budget is met. Step 3 chooses the subset that best fits the current window, assigns those to $D\text{-}DMR_{intra}$ (or $D\text{-}DMR_{inter}$), and continues to the next scheduling point.

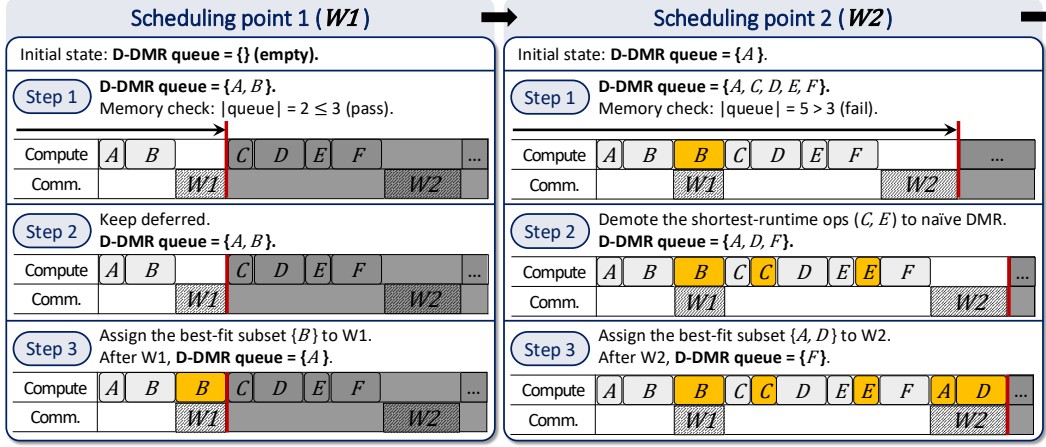

Figure 10: Fine-grained (i.e., per-operation) planning for D-DMR of static DMR planner.

Figure 10 illustrates this process, assuming a memory budget equal to the input/output tensor size of three operations, with all operations ($A$–$F$) initially unassigned and all communication windows ($W1$ and $W2$) unused. At the first scheduling point ($W1$), no demotion occurs since the memory budget is not violated, and $B$ is assigned to $W1$. At the second scheduling point ($W2$), the memory budget is exceeded, so $C$ and $E$ are demoted to *Naïve-DMR*, and $A$ and $D$ are assigned to $W2$.

**For MoE Models**. For MoE models, the dynamic planner—while less efficient than the static planner (see Appendix C.1 for details)—performs Phase 3 at runtime. When executing primary execu-

tions of operations that are not yet assigned, the dynamic planner enqueues their inputs and outputs into the D-DMR queue. Upon encountering a communication window, the planner uses it as follows: for static windows (e.g., TP), it selects a best-fit subset from the queue and launches those operations; for dynamic windows (e.g., EP), it dequeues and executes operations one by one until the window closes or the queue empties. However, the queue cannot grow indefinitely. The dynamic planner continuously monitors memory usage, and if usage nears the limit, it immediately dequeues and executes checker operations (i.e., falls back to *Naïve-DMR*) to satisfy the memory constraint.

## 6 EVALUATION

We implement SpareTrain built on PyTorch 2.9, extending its training stack with new error detection capabilities to support DMR verification during training. Section 6.1 details the evaluation training setup; Section 6.2 presents end-to-end throughput results; and Section 6.3 quantifies the contribution of each component via ablation studies.

In addition, Appendix C.2 describes our implementation decision enabling effective D-DMR$_{intra}$, and Appendix D provides further experimental setup details. As part of our extended experiments, Appendix E.1 presents a sensitivity study on sequence length; Appendix E.2 presents an evaluation of SpareTrain combined with asyncTP, which also leverages GPU idle time in TP; and Appendix E.3 explores alternative parallelism strategies for MoE models.

### 6.1 SETUP

**Hardware Configurations**. We conduct experiments on a GPU cluster with up to four nodes. Each node is equipped with 8 NVIDIA H200-SXM5 GPUs (141GB memory per GPU). Inter-node communication uses InfiniBand (per node: $8 \times 400$Gb/s links) with GPUDirect RDMA enabled, while intra-node GPU–GPU communication leverages NVSwitch with 900GB/s bandwidth.

To evaluate SpareTrain under varying memory budgets (i.e., different GPU memory capacities), we emulate 80GB and 94GB configurations by capping each GPU's available memory. These settings correspond to the two NVIDIA H100-SXM5 variants, another widely adopted GPU line for LLM training. Because H100 and H200 share the same compute capabilities and cache hierarchy and differ primarily in memory specifications, this emulation reasonably reflects practical deployments on H100 GPUs.

**Model and Training Configurations**. We evaluate SpareTrain on both dense and MoE models. For dense models we use Llama-3-70B and Mistral-Large (123B). For the MoE model we use Llama-4-Scout (109B). Dense models are trained using a combination of TP and PP. For the MoE model, expert layers use EP and non-expert layers use TP. TP and EP are confined within a node and set to degree eight (matching the eight GPUs per node), while PP spans across nodes. To determine practical setups, we sweep the PP degree (up to four) and the AC degree, selecting the configuration that achieves the highest throughput for each model. All training runs use the TorchTitan framework (Liang et al., 2025a) with mixed-precision training (Micikevicius et al., 2018), and enable `torch.compile` across all setups.

### 6.2 TRAINING THROUGHPUT

Figure 11 presents training throughput, measured in tokens per second, of *No-DMR* (a vanilla training system without DMR), *Naïve-Only* (naïve DMR applied to all operations), and SpareTrain across different memory capacities and batch sizes. We highlight key observations from the results below.

**Dense Models (Llama-3-70B and Mistral-Large)**. For Llama-3-70B, averaged over the evaluated batch sizes, SpareTrain achieves 32%, 31%, and 33% higher throughput than *Naïve-Only* at 80, 94, and 141GB, respectively, while trailing *No-DMR* by 3.3%, 13.8%, and 10.9%. The relatively smaller gap at 80GB arises because this configuration requires a higher PP degree ($PP$=3, compared to $PP$=2 at 94/141GB) due to memory constraints. The increased PP degree expands communication windows, which in turn allows for more aggressive D-DMR$_{inter}$. The same trend is observed for Mistral-Large. Averaged over the evaluated batch sizes, SpareTrain improves throughput by 26%, 21%, and 23% over *Naïve-Only*, while incurring slowdowns of 6.1%, 11.0%, and 9.8% relative

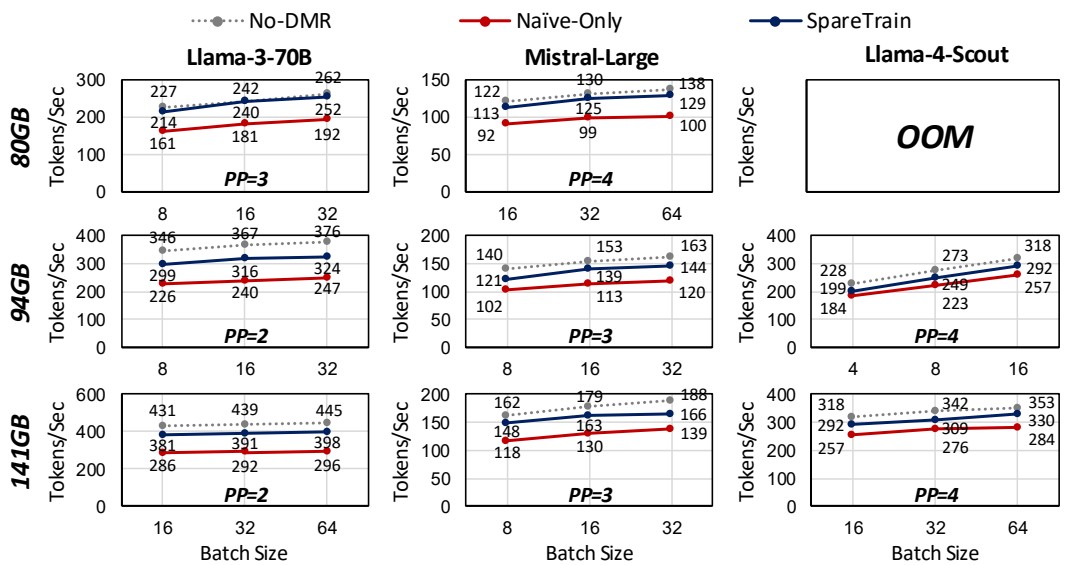

Figure 11: Training throughput of *No-DMR*, *Naïve-Only* and SpareTrain.

to *No-DMR*, respectively. In summary, SpareTrain accelerates DMR-protected training by roughly 30%. Put differently, it enables full DMR with only about a 10% slowdown.

**MoE Model (Llama-4-Scout).** For Llama-4-Scout, averaged over the evaluated batch sizes, Spare-Train improves throughput over *Naïve-Only* by 11% and 14% at 94GB and 141GB, while incurring slowdowns of 9.8% and 8.1% relative to *No-DMR*, respectively. The 80GB setup is excluded due to out-of-memory (OOM) errors. The performance gaps among *Naïve-Only*, SpareTrain, and *No-DMR* are relatively smaller for MoE models than for dense models. This is because the relative DMR overhead is lower: MoE models incur higher communication costs relative to computation, thus DMR contributes less to the overall runtime. Nevertheless, SpareTrain still effectively reduces DMR overhead.

## 6.3 ABLATION STUDY

Figure 12 demonstrates how each phase of DMR planning contributes to training throughput improvement. For the dense model (Mistral-Large), we progressively enable the three phases of the static DMR planner, and for the MoE model (Llama-4-Scout), we instead enable Phases 1 and 2 of the static planner followed by the dynamic planner. All three phases significantly contribute to improving training throughput. This implies that both P-DMR and D-DMR play critical roles, and that within D-DMR, both fine-grained and coarse-grained planning are essential.

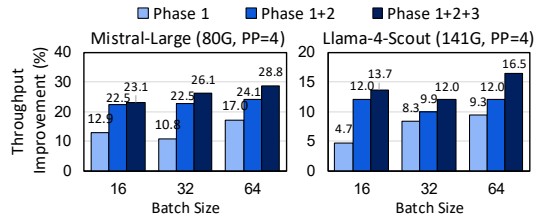

Figure 12: Ablation study of the three phases of DMR planning in SpareTrain.

## 7 CONCLUSION

In this paper, we present SpareTrain, a novel approach to achieving exact silent data corruption detection in LLM training with minimal overhead. By leveraging the inherent redundancy in activation checkpointing and GPU under-utilization, our system demonstrates that DMR can be practically applied without prohibitive costs. As LLMs continue to scale to larger models, SpareTrain enables reliable training while maintaining computational efficiency, bridging the gap between theoretical fault tolerance mechanisms and practical deployment requirements.

ACKNOWLEDGMENTS

This work was supported by Samsung Electronics Co., Ltd. (Device Solutions) and the Institute of Information & Communications Technology Planning & Evaluation (IITP) grant funded by the Korea government (MSIT) (RS-2025-02217656). Jae W. Lee is the corresponding author.

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

## A  NEAR-ZERO MEMORY OVERHEAD OF P-DMR

P-DMR requires storing the outputs of each AC segment's primary execution for later comparison. In practice, this rarely introduces additional memory overhead. If an AC segment is followed by another AC segment within the same stage, its outputs are already needed as checkpoint inputs for the next segment. Conversely, if the AC segment is followed by operations not covered by activation checkpointing, the outputs are naturally preserved in memory as inputs to those operations (since those operations retain activations for the backward pass). Thus, in most cases, P-DMR simply reuses tensors that are already in memory.

The only exception arises when the last operation of an AC segment is also the last operation of a pipeline stage. Since no subsequent operation consumes these outputs, they must be explicitly retained until the backward pass recomputes them for comparison. The worst-case additional retention depends on the pipeline schedule and pipeline degree: when a stage processes multiple forward microbatches consecutively before their backward passes, it must keep the last outputs of all those microbatches concurrently.

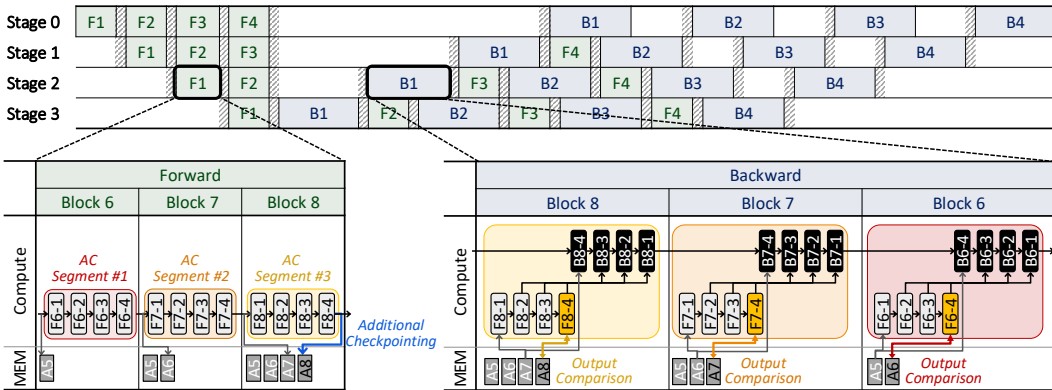

Figure 13: Illustration of P-DMR memory retention under a 1F1B pipeline schedule when a 12-block transformer model is partitioned across four pipeline stages ($PP = 4$). The figure highlights the additional memory required by P-DMR, where only the final forward stage outputs can introduce any additional memory.

Figure 13 illustrates this scenario on a 12-block transformer model partitioned into four pipeline stages (three blocks per stage) under a 1F1B schedule. For simplicity, we assume AC is applied at the granularity of each block. Regardless of the specific AC segmentation, the only tensor that can introduce additional memory overhead is the last output of each stage (e.g., A8 for Stage 2). In this example, Stage 0 processes four consecutive forward microbatches (F1–F4), requiring additional memory to store four sets of outputs until their corresponding backward passes.

Formally, the extra per-GPU retention is proportional to the product of bytes per element, sequence length, hidden dimension, the number of stages per node, the maximum number of consecutive forwards, and the microbatch size, all divided by the TP degree. For example, on Llama-3-70B (sequence length 8192, hidden 8192, microbatch size 1, TP=8), a 1F1B schedule with pipeline degree $PP$=3 adds at most 96MB per GPU. Even with Interleaved-1F1B (two stages per node), the overhead is at most 192MB per GPU—negligible compared to modern GPU capacities (80–141 GB)—and did not affect the actual peak memory usage.

In the backward pass, P-DMR may marginally extend the lifetime of certain tensors. As illustrated in Figure 13, A7 would be freed immediately after the backward of block 8 in standard AC, but must be retained until the backward of block 7 to enable comparison. This effect does not accumulate across AC segments and persists only for a short duration, making the additional retention negligible in practice.

## B AGC SEGMENT SELECTION RESULT

In Phase 2, the AGC segment selection results for Llama-3-70B are summarized in Table 1, covering device memory budgets of 80GB, 94GB, and 141GB.

Table 1: Phase 2 evaluation results for Llama-3-70B training, showing the actual constraints and the selected AGC segments. Memory headroom and Memory cost are denoted in MB, while PP slack, Execution time, and Time saved are denoted in ms.

| Memory Budget | Batch Size | PP Degree | PP Stage | # Blocks | Constraints (Per block) | | Selected Segment (Per block) | | |
| --- | --- | --- | --- | --- | --- | --- | --- | --- | --- |
| | | | | | Memory headroom | PP slack | Memory cost | Execution time | Time saved |
| 80GB | 32 | $PP=3$ | 0 | 26 | 706 | 26.602 | 472 | 15.627 | 7.860 |
| | | | 1 | 27 | 853 | 13.229 | 104 | 13.178 | 5.441 |
| | | | 2 | 27 | 380 | 25.020 | 136 | 13.360 | 5.659 |
| 94GB | 32 | $PP=2$ | 0 | 40 | 257 | 10.845 | 120 | 6.939 | 3.484 |
| | | | 1 | 40 | 84 | 10.507 | 80 | 6.617 | 3.747 |
| 141GB | 64 | $PP=2$ | 0 | 40 | 612 | 10.774 | 400 | 8.166 | 4.348 |
| | | | 1 | 40 | 715 | 10.398 | 400 | 8.146 | 4.372 |

Comparing the 80GB setting with the larger budgets, a larger memory budget allows a lower PP degree, which reduces per-block PP slack because inter-node communication is divided among more transformer blocks per stage. For example, each stage hosts 26–27 blocks at $PP=3$ but about 40 blocks at $PP=2$; consequently, the available PP communication slack must be distributed across more blocks, shrinking the per-block slack. This, in turn, leads the planner to favor smaller segments when the PP degree is lower.

Between the 94GB and 141GB settings, the pipeline degree remains the same, so the per-block PP slack is nearly identical. However, the additional memory headroom at 141GB allows for larger segments to be selected, whereas the 94GB case is restricted to smaller ones.

Overall, 41–57% of the segment execution time yields actual time saved, since intra-segment communication and redundant operations already protected by P-DMR still need to be re-executed. Nevertheless, by selecting efficient segments that respect both memory headroom and PP slack, the planner effectively utilizes idle communication windows by deferring DMR work off to the critical path.

## C OVERLAPPING COMMUNICATION AND COMPUTATION KERNELS

### C.1 D-DMR$_{intra}$ IN PHASE 3: STATIC VS. DYNAMIC PLANNING

When performing D-DMR$_{intra}$, two main inefficiencies arise under dynamic planning, where certain communication durations are not known in advance.

The first is the launch overhead in dynamic communication. In static communication, operations executed under D-DMR$_{intra}$ can leverage predetermined communication durations to launch all best-fit operations from the queue simultaneously at the start of communication. However, in dynamic communication, operations under D-DMR$_{intra}$ are launched one by one until communication completes, incurring repeated overhead from kernel launches and communication completion checks. Figure 14 illustrates this difference between static and dynamic communication.

Second, naïve DMR in dynamic planning requires longer memory retention than in static planning. In static planning, operations designated for naïve DMR are predetermined as *Naïve-DMR* during the offline phase, allowing them to be executed twice back-to-back on the critical path. In dynamic planning, operations are opportunistically placed in the D-DMR queue first, and later executed with naïve DMR on the critical path whenever memory resources are tight or the communication window is not long enough to allow overlap. Although the critical path execution remains the same, memory retention is prolonged because operations cannot be processed immediately due to the uncertainty of whether each operation will eventually be executed with naïve DMR.

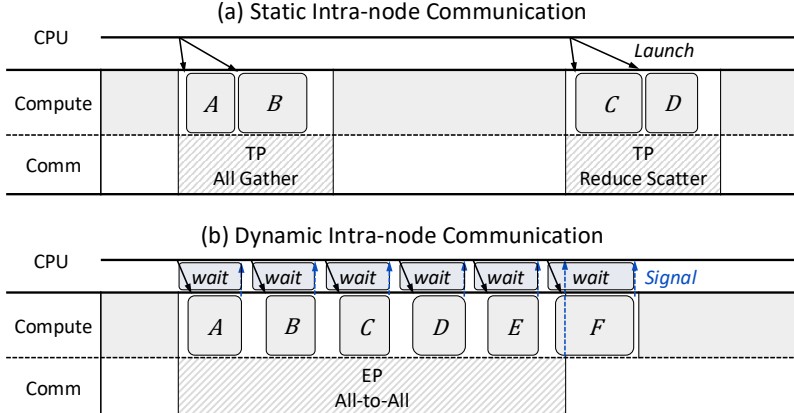

Figure 14: Overlap of checker executions with intra-node communication in (a) static and (b) dynamic communication.

## C.2 INTRA-NODE COMMUNICATION VIA NCCL VS. P2P

Intra-node collective communication is typically performed using collective libraries such as NCCL. However, NCCL collective operations are implemented as device kernels that consume SM (the GPU's compute cores), leading to contention with concurrent compute kernels during overlap. As a result, although computation–communication overlap appears feasible in theory, the performance gains are often minimal in practice.

In contrast, peer-to-peer (P2P) transfers operate through NVLink using dedicated copy engines (DMA) and do not consume SM resources. This substantially reduces resource contention and enables stable overlap. PyTorch's symmetric_memory, which implements P2P-based collective communication, may be slightly slower than highly optimized NCCL kernels, but its clear ability to overlap leads to higher overall throughput than sequential execution of computation and communication. Therefore, when SpareTrain exploits intra-node communication windows for D-DMR$_{intra}$, we rely on PyTorch's symmetric_memory instead of NCCL to achieve effective overlap.

## D EXPERIMENTAL SETUP DETAILS

### D.1 HARDWARE CONFIGURATION AND MEMORY CAPACITY EMULATION

Detailed hardware specifications as shown in Table 2. We evaluate SpareTrain under varying memory budgets (or GPU clusters with different memory capacities). To emulate 80 GB and 94 GB configurations, we cap each GPU's usable memory by setting an upper bound on all reserved memory (allocated and unallocated) managed by PyTorch's `CachingAllocator` (PyTorch Team, 2025).

Table 2: Hardware specifications of a single evaluation node.

| Category | Specification |
|---|---|
| CPU | 2 × Intel Xeon Platinum 8580 (128 cores) |
| System Memory | 2,048 GB DDR5-5600 |
| GPU | 8 × NVIDIA H200-SXM5 |
| GPU Memory | 141 GB HBM3e per GPU |
| Inter-node Network | 8 × 400 Gb/s InfiniBand (RDMA) |
| Intra-node (GPU P2P) | NVSwitch, 900 GB/s |

## D.2 Configuring Pipeline Parallelism and Activation Checkpointing Degrees

While *full* activation checkpointing stores only layer inputs and recomputes all intermediate activations, *selective* strategies provide finer control for better memory-computation trade-offs (He & Yu, 2023; Korthikanti et al., 2023; Labatut, 2025). PyTorch's `torch.compile` supports automated selective checkpointing through the Memory Budget API (Labatut, 2025). Given `ac_budget` $\in [0,1]$—where 0 enforces full checkpointing and 1 disables it—the compiler automatically finds pareto-optimal checkpointing plans.

Table 3: Training performance of Llama-3-70B on 16 H200 GPUs (141GB) under varying `ac_budget` settings.

| `ac_budget` | 0.0 | 0.25 | 0.5 | 0.75 | 1.0 |
|---|---|---|---|---|---|
| Tokens/Sec | 376.3 | 421.9 | 438.3 | 439.0 | 436.5 |
| Peak Memory (GB) | 82.71 | 101.47 | 107.17 | 107.17 | 129.39 |

The performance implications of activation checkpointing, however, are nontrivial. While tighter memory budgets enforce high degree of checkpointing, relaxing the budget does not always yield higher throughput. For example, Table 3 shows training performance of Llama-3-70B on 16 H200 GPUs under $PP = 2, TP = 8$: although `ac_budget` = 1.0 consumes the most memory (129.39 GB), throughput peaks at `ac_budget` = 0.75, which uses less memory (107.17 GB). This highlights that the optimal degree of checkpointing cannot be derived analytically and must be identified empirically through sweeps over `ac_budget`.

Similarly, pipeline parallelism degree ($PP$) creates trade-offs: higher PP degree reduces per-device memory pressure but introduces communication overhead. The combined effect of PP degree and activation checkpointing degree means optimal configurations require joint empirical evaluation of both parameters to achieve best throughput under given hardware constraints.

To ensure fair comparison, we establish strong baselines by sweeping all feasible PP degrees (up to 4 nodes) and `ac_budget` values {0, 0.25, 0.5, 0.75, 1.0} for baseline (*No-DMR*), selecting the highest-throughput configuration. For SpareTrain, we use the same PP degree but independently sweep `ac_budget` values, since `ac_budget` directly determines both the extent of P-DMR coverage and the available memory headroom for D-DMR execution. This sweep is also required for vanilla training, so our method incurs no extra search overhead.

In practice, using the minimum PP degree that fits the model consistently outperformed configurations with higher PP degrees and relaxed `ac_budget` values; therefore, the selected PP degrees are shown in Figure 11 in Section 6. For `ac_budget`, however, the optimal value can differ: for Llama-3-70B with a 141GB memory budget and batch size of 64, the vanilla baseline achieved its best performance with `ac_budget` = 1.0 (no activation checkpointing), whereas SpareTrain performed optimally at `ac_budget` = 0.5, which provides better utilization of both P-DMR and D-DMR by reserving additional memory headroom.

## E Extended Evaluation

### E.1 Effect of Sequence Length on SpareTrain

Figure 15 shows the impact of sequence length (4K, 8K, and 16K tokens per sample) on the training throughput of SpareTrain. Experiments were conducted on Mistral-Large ($PP=4, TP=8$) using 32 GPUs, each with 94GB of memory. Averaged over batch sizes, the throughput of SpareTrain is lower than that of *No-DMR* by 4.6%, 6.1%, and 3.1% at 4K, 8K, and 16K, respectively. These results indicate that the relative overhead of SpareTrain remains small and consistent, suggesting that its effectiveness is largely unaffected by sequence length.

### E.2 SpareTrain Performance under Communication–Computation Overlap

To mitigate the communication overhead, numerous techniques have been proposed to overlap TP communication with computation (Wang et al., 2022; Jangda et al., 2022). The key idea in these

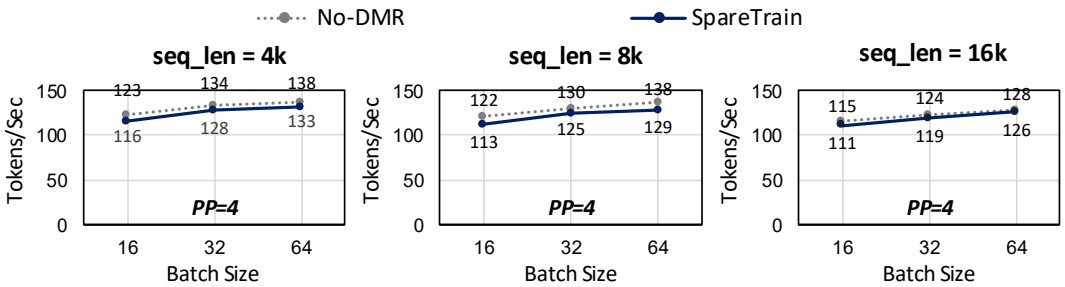

Figure 15: Training throughput of No-DMR and SpareTrain across varying sequence length.

approaches is to decompose both communication and computation into finer-grained steps so that they can proceed in a pipelined fashion. A representative implementation is PyTorch's async tensor parallelism (`asyncTP`), which overlaps TP communication with its dependent GEMM computation. Specifically, `asyncTP` breaks down collective operations (e.g., all-gather, reduce-scatter) into chunked send/recv steps (implemented via P2P communication with symmetric_memory) and splits the subsequent matrix multiplication into sub-matmul kernels. This design allows each sub-matmul to run in parallel with the transfer of the next communication chunk, achieving effective overlap.

We evaluate SpareTrain under this overlap mechanism by training Mistral-Large ($PP = 4$, 80GB memory per GPU) with `asyncTP` enabled. Table 4 reports throughput for No-DMR and Spare-Train under different batch sizes, with results in parentheses showing the corresponding runs without `asyncTP`. On average, SpareTrain incurs only a 4.2% slowdown relative to No-DMR with `asyncTP`.

Table 4: Training throughput comparison between No-DMR and SpareTrain with `asyncTP` enabled under different batch sizes, evaluated on Mistral-Large. Values in parentheses show the corresponding results without `asyncTP`.

|  | Batch Size | | |
|  | 16 | 32 | 64 |
| --- | --- | --- | --- |
| No-DMR | 123.6 (121.8) | 132.1 (130.0) | 140.5 (137.5) |
| SpareTrain | 118.5 (112.6) | 126.2 (124.5) | 134.9 (128.6) |
| **Slowdown** | **4.2%** (7.6%) | **4.5%** (4.3%) | **4.0%** (6.5%) |

Notably, the slowdown of SpareTrain can be smaller with `asyncTP` enabled. Without `asyncTP`, only SpareTrain suffers from communication performance degradation due to switching from NCCL to P2P-based communication for D-DMR$_{intra}$ operations, while the *No-DMR* continues using optimized NCCL collectives. However, `asyncTP` internally replaces NCCL collectives with P2P-based communication via symmetric_memory for both systems, eliminating this communication implementation disparity. Although P2P communication takes slightly longer than NCCL in communication time, this change does not benefit *No-DMR*; however, SpareTrain can leverage it as additional slack for D-DMR$_{intra}$ operations.

Therefore, in some cases, enabling `asyncTP` can even reduce the relative performance gap between SpareTrain and the *No-DMR*. Overall, these results demonstrate that SpareTrain remains compatible with existing overlap mechanisms.

### E.3 ALTERNATIVE PARALLELISM FOR MoE MODEL

Context Parallelism (CP) shards the input sequence along the sequence dimension across GPUs (Liu et al., 2023a). By splitting long sequences, CP reduces per-GPU memory usage and enables training with larger sequence lengths. However, unlike sequence-invariant operations (e.g., feed-forward networks, normalization), self-attention requires the full sequence context, which incurs additional communication of key/value tensors across GPUs.

For MoE training, the commonly recommended configuration combines EP for MoE layers with either TP or CP for non-MoE layers (Liu et al., 2025; Jin et al., 2025). Our main evaluation adopted TP for non-MoE layers, as it provided better throughput. To validate the generality of our approach, we also conducted experiments using CP for non-MoE layers while retaining EP for MoE layers. In these experiments, we set $PP=4$ across nodes, while non-MoE layers used CP with degree $8$ and MoE layers used EP with degree $8$. As in standard practice, these intra-node parallelism dimensions (CP, TP, and EP) matched the number of GPUs per node to mitigate the cost of frequent communication.

Table 5 reports the throughput under this configuration. The results confirm that SpareTrain maintains complete DMR coverage with only reasonable slowdowns, demonstrating that our technique is effective even when CP replaces TP in MoE training.

Table 5: Throughput comparison for MoE training when using CP for non-MoE layers and EP for MoE layers. Results are for Llama-4-Scout with a 141GB GPU memory budget and $PP=4$.

| | Batch Size | | |
| --- | --- | --- | --- |
| | 4 | 8 | 16 |
| No-DMR | 160.1 | 229.1 | 296.4 |
| SpareTrain | 152.1 | 222.5 | 284.1 |
| **Slowdown** | **5.0%** | **2.9%** | **4.2%** |

### E.4 P-DMR Coverage and Exception Cases Under Varying Activation Checkpointing Configurations

The coverage of P-DMR depends directly on the degree of activation checkpointing (AC), which determines how many operations are recomputed during the backward pass. As described in Appendix D.2, the `ac_budget` parameter of `torch.compile` provides fine-grained operation-level activation checkpointing. We analyze both P-DMR coverage for varying `ac_budget` and the frequency of exception cases in the end-to-end training experiments from Section 6.

Table 6 shows P-DMR coverage for varying `ac_budget` values, measured on Llama-3-70B trained on H200 GPUs (141GB) with TP=8. Both rows represent percentages of total forward operation execution time: the first row shows the fraction spent on recomputed operations, and the second row shows the fraction covered by P-DMR. The coverage closely tracks the recomputation ratio. At `ac_budget` = 0.0 (full AC), nearly all forward operations are recomputed and thus verified by P-DMR. As `ac_budget` increases, P-DMR coverage proportionally reduces.

Table 6: Recomputation ratio and P-DMR coverage for Llama-3-70B under varying `ac_budget`. Values are percentages of total forward operation execution time.

| `ac_budget` | 0.0 | 0.25 | 0.5 | 0.75 | 1.0 |
| --- | --- | --- | --- | --- | --- |
| Recomputed Operations (%) | 99.1 | 55.8 | 22.7 | 22.7 | 0.4 |
| Operations Covered by P-DMR (%) | 99.1 | 50.9 | 21.3 | 21.3 | 0.4 |

As described in Section 5.1, Phase 1 applies P-DMR by default to all operations recomputed by AC. However, an exception case occurs when the last operation of an AC segment requires expensive communication to gather its input operands—in such cases, naïve DMR becomes preferable. Table 7 reports the exception case ratio observed during the training runs from Section 6 for Llama-3-70B.

Overall, exception cases either do not occur or appear at a low ratio (under 20%), highlighting the practical effectiveness of P-DMR. The results correlate with `ac_budget`: at 80GB and 94GB memory budgets, `ac_budget` = 0.0 enforces full activation checkpointing, where each transformer layer forms a single large AC segment, so no exception cases occur. Exception cases appear only at higher memory budgets (e.g., 141GB with `ac_budget` = 0.5), where selective activation checkpointing creates smaller, fine-grained segments. In these smaller segments, communication costs can more easily outweigh recomputation costs, occasionally triggering the exception condition. Importantly, even at

an 18.2% exception ratio, the actual performance impact is smaller still, as shown by the gap between recomputed operations and P-DMR coverage in Table 6—exception cases inherently involve segments with low computational cost.

Table 7: Exception case ratio in Phase 1 (P-DMR) for Llama-3-70B across memory budgets.

| Memory Budget | 80GB | 94GB | 141GB |
|---|---|---|---|
| Exception Case Ratio | 0.0% (0/80) | 0.0% (0/80) | 18.2% (160/880) |

