# OpenReview forum: "SpareTrain: Fault-Tolerant LLM Training via Low-Cost Dual Modular Redundancy"
_ICLR.cc/2026/Conference — ICLR 2026 Poster_

### Official Review · Reviewer_EkeY · 2025-10-27

**Soundness:** 3
**Presentation:** 2
**Contribution:** 2
**Rating:** 6
**Confidence:** 2

**Summary:**

The paper addresses silent data corruption detection by integrating Piggyback-DMR and Deferred-DMR into a novel framework called SpareTrain.

**Strengths:**

1. The proposed method is applied to high-performance GPUs in a large-scale cluster.
2. The proposed method can be applied to both MoE and non-MoE models.
3. The proposed method introduces only a 3–14% overhead compared to baseline training without SDC protection.

**Weaknesses:**

1. D-DMR appears to work only with pipeline parallelism, which may limit its applicability.
2. It seems that the proposed method relies on high-performance GPUs, where computation time is significantly faster than communication time. For GPUs like the H20, the effectiveness of the proposed method remains uncertain.
3. We seem to observe a contradiction: we use pipeline parallelism due to memory constraints, yet the proposed method requires additional memory. Does this imply that, with given memory, the baseline could use smaller pipeline parallelism to achieve higher throughput? This seems like a better baseline.

**Questions:**

In Piggyback-DMR, could some internal SDCs be overlooked, since it only checks the input and output of Activation Gradient Checkpointing?

---

> ### Author Response · Authors · 2025-11-21
>
> We thank the reviewer for the valuable feedback. We respond to individual points from your review below.
>
> > 1. D-DMR appears to work only with pipeline parallelism, which may limit its applicability.
> >
>
> To clarify, D-DMR is not limited to pipeline parallelism; it can leverage any form of communication-induced idle time. Even when pipeline parallelism is not used, D-DMR remains applicable by utilizing idle periods created by other parallelism strategies such as tensor parallelism or expert parallelism. Meanwhile, our primary target is large models with tens or hundreds of billions, if not trillions, of parameters, and for such models, pipeline parallelism has become the de facto standard in practice [1-6].
>
> [1] Chu, Weiwei, et al. "Scaling Llama 3 Training with Efficient Parallelism Strategies." Proceedings of the 52nd Annual International Symposium on Computer Architecture. 2025.
>
> [2] Liu, Dennis, et al. "MoE Parallel Folding: Heterogeneous Parallelism Mappings for Efficient Large-Scale MoE Model Training with Megatron Core." arXiv:2504.14960 (2025).
>
> [3] Jin, C., et al. "Megascale-moe: Large-scale communication-efficient training of mixture-of-experts models in production.” EuroSys (2026).
>
> [4] Adler, Bo, et al. "Nemotron-4 340b technical report." arXiv:2406.11704 (2024).
>
> [5] Almazrouei, Ebtesam, et al. "The falcon series of open language models." arXiv:2311.16867 (2023).
>
> [6] Dubey, A., et al. "The Llama 3 Herd of Models." arXiv:2407.21783 (2024).
>
> > 2. It seems that the proposed method relies on high-performance GPUs, where computation time is significantly faster than communication time. For GPUs like the H20, the effectiveness of the proposed method remains uncertain.
> >
>
> It is true that for GPUs with more limited compute capability, such as the H20 you mentioned, the effectiveness of SpareTrain may be reduced because the relative ratio of computation time to communication time becomes smaller.  However, such GPUs are generally not used for large-scale LLM training in practice; they are typically deployed for inference rather than training. Production-grade LLM training relies on training-optimized GPUs with high compute capabilities such as H100 [1-6], H200 [7], or H800 [8-10]. Our evaluation on H200 directly represents mainstream training hardware used in real-world settings.
>
> [1] Dubey, A., et al. "The Llama 3 Herd of Models." arXiv:2407.21783 (2024).
>
> [2] Adler, Bo, et al. "Nemotron-4 340b technical report." arXiv:2406.11704 (2024).
>
> [3] Agarwal, S., et al. "gpt-oss-120b & gpt-oss-20b model card." arXiv:2508.10925 (2025).
>
> [4] Abdin, M., et al. "Phi-4-reasoning technical report." arXiv:2504.21318 (2025).
>
> [5] Han, S., et al. "Trillion 7B Technical Report." arXiv:2504.15431 (2025).
>
> [6] MosaicML. "Introducing MPT-30B." https://www.databricks.com/blog/mpt-30b
>
> [7] Dade, N.O., Rahat, M.H. "Litespark Technical Report." arXiv:2510.02483 (2025).
>
> [8] Liu, A., et al. "Deepseek-v2." arXiv:2405.04434 (2024).
>
> [9] DeepSeek-AI. "DeepSeek-V3 Technical Report." arXiv:2412.19437 (2024).
>
> [10] Jin, C., et al. "Megascale-moe: Large-scale communication-efficient training of mixture-of-experts models in production.” EuroSys (2026).
>
> > 3. We seem to observe a contradiction: we use pipeline parallelism due to memory constraints, yet the proposed method requires additional memory. Does this imply that, with given memory, the baseline could use smaller pipeline parallelism to achieve higher throughput? This seems like a better baseline.
> >
>
> We have already used the smallest feasible pipeline parallelism (PP) degree for the baseline. As described in Appendix D.2, we sweep both the PP degree and the activation checkpointing degree and select the configuration that yields the highest throughput, ensuring that the baseline reflects the best configuration attainable under the given memory constraints. As you also inferred, a smaller PP degree consistently achieves higher throughput, and therefore we adopt that configuration for all evaluations. Given this best-effort baseline, SpareTrain only utilizes the remaining memory capacity, which would otherwise remain unused.

---

> > ### Author Response · Authors · 2025-11-21
> >
> > Additionally, we provide responses to the questions.
> >
> > > In Piggyback-DMR, could some internal SDCs be overlooked, since it only checks the input and output of Activation Gradient Checkpointing?
> > >
> >
> > Piggyback-DMR does not overlook internal SDCs within an activation-checkpointing (AC) segment. In AC, the entire forward segment is fully recomputed during the backward pass, and P-DMR compares the outputs of the primary and checker executions. Because every internal operation (within a AC segment) contributes to this final output, any SDC in an intermediate step will propagate to the segment output and be detected during comparison (under the same standard fault model assumed by traditional DMR [1,2,3]). Operations that fall outside AC segments are still protected separately through D-DMR or naïve DMR, ensuring full coverage across the entire model.
> >
> > [1] Reinhardt, Steven K., and Shubhendu S. Mukherjee. "Transient fault detection via simultaneous multithreading." Proceedings of the 27th annual international symposium on Computer architecture. 2000.
> >
> > [2] Mukherjee, Shubhendu S., Michael Kontz, and Steven K. Reinhardt. "Detailed design and evaluation of redundant multithreading alternatives." ACM SIGARCH Computer Architecture News 30.2 (2002): 99-110.
> >
> > [3] Jeon, Hyeran, and Murali Annavaram. "Warped-DMR: Light-weight error detection for GPGPU." 2012 45th Annual IEEE/ACM International Symposium on Microarchitecture. IEEE, 2012.

---

> > > ### Comment · Reviewer_EkeY · 2025-11-25
> > >
> > > Thank you so much for your detailed and helpful explanation. You mentioned that "any SDC in an intermediate step will propagate to the segment output and be detected during comparison." I was wondering if we could therefore conclude that the design inherently prevents any fault masking from occurring within a single AC segment? For example, is there any conceivable scenario where two independent faults might occur and their effects somehow cancel each other out, leading to a deceptively correct segment output that would pass the final comparison undetected?

---

> > > > ### Author Response · Authors · 2025-11-28
> > > >
> > > > Short answer is yes. In the corner cases you described, DMR performed at the AC segment level may result in undetected SDC. Eliminating such cases would require DMR at per-operation granularity rather than per-segment. However, given the extremely low likelihood of such fault-masking events, we consider this risk negligible and regard our scheme as complete in terms of SDC detection.
> > > > This is analogous to the standard assumption behind conventional DMR systems, where we typically ignore the exceedingly small chance that the primary and checker executions fail in the same way and thus allow an error to slip through. With such risks considered negligible, DMR is usually regarded as offering full detection coverage in practice.

---

### Official Review · Reviewer_MpZL · 2025-10-29

**Soundness:** 3
**Presentation:** 2
**Contribution:** 3
**Rating:** 6
**Confidence:** 4

**Summary:**

This paper presents SpareTrain, a system for fault-tolerant LLM training that reduces the cost of dual modular redundancy (DMR). It introduces two mechanisms: Piggyback-DMR (P-DMR), which reuses activation recomputation from checkpointing for validation, and Deferred-DMR (D-DMR), which exploits idle GPU time (e.g., pipeline bubbles) to execute redundant checks. A planner coordinates these modes to balance reliability and efficiency. Experiments on up to 32 H200 NPUs show up to 35% speedup over naïve DMR while maintaining full coverage claims. However, the paper provides limited validation of correctness and scalability.

**Strengths:**

- Addresses an important reliability problem in large-scale LLM training, where silent hardware errors can silently corrupt long-running training.
- The idea of reusing existing computation (recompute or idle slots) for redundancy is intuitive and practically relevant, reducing the prohibitive cost of full DMR.
- The paper demonstrates a non-trivial engineering effort, with experiments on real Ascend NPU clusters up to 8192 devices.
- The proposed planner that automatically assigns P-DMR, D-DMR, or Full-DMR modes based on profiling data is conceptually elegant and system-aware.
- Results show that SpareTrain can significantly lower redundancy overhead while maintaining theoretical detection coverage.

**Weaknesses:**

Although the paper presents a conceptually reasonable design, the overall analytical depth and experimental coverage are limited, leading to weak interpretability of the proposed DMR planner.

First, the relationship between the DMR planning mechanism and the underlying parallel configuration (such as tensor, pipeline, and expert parallelism) is never formally analyzed.
The effectiveness of both P-DMR and D-DMR clearly depends on these configurations—pipeline depth determines bubble size, tensor-parallel degree affects collective communication cost, and expert parallelism introduces dynamic execution patterns.
However, the paper only illustrates a few heuristic cases and does not provide any mathematical modeling, cost function, or decision boundary describing when a mode switch (e.g., from P-DMR to naïve DMR) should occur.
As a result, the planner behaves like a hand-tuned heuristic rather than a principled optimization framework.

Second, the interaction between checkpointing frequency and P-DMR coverage is completely unexamined.
Since activation checkpointing recomputes every n layers, the interval n directly controls both recomputation cost and P-DMR reuse potential.
Without quantitative analysis of how n influences memory usage, recomputation overlap, or fault detection coverage, it is unclear how the proposed planner would adapt under different checkpoint configurations.

Third, the experimental evaluation is overly high-level.
It reports only end-to-end throughput and speedup ratios, without any breakdown of how many operators are assigned to P-DMR, D-DMR, or naïve DMR modes under different parallel setups.
There is no analysis of idle-window utilization, checkpoint overlap ratio, or the planner’s sensitivity to communication intensity.
Consequently, the claimed throughput improvement lacks interpretability and cannot be correlated with the underlying design decisions.

Finally, the planner’s objective and constraints are only described qualitatively (“maximize reuse given time/memory slack”) without any explicit cost model or optimization formulation.
This makes it difficult to reason about convergence, scalability, or generalization to unseen workloads.

**Questions:**

- How does the proposed planner adapt to different parallel configurations (e.g., TP, PP, EP) that directly affect communication cost and idle-window size?
- How does the checkpointing frequency (n) influence P-DMR coverage and overall verification cost?
- Is there a quantitative cost model or optimization formulation guiding the planner’s decisions, or are all heuristics empirically tuned?
- Can the authors provide the distribution of P-DMR, D-DMR, and naïve DMR operators under different configurations to explain the observed throughput changes?

---

> ### Author Response · Authors · 2025-11-21
>
> We are grateful for the reviewer's constructive feedback. We provide responses to the specific points from your review below.
>
> > Q1. How does the proposed planner adapt to different parallel configurations (e.g., TP, PP, EP) that directly affect communication cost and idle-window size?
> >
>
> We clarify that SpareTrain’s planner implementation is not tied to any specific parallelism configuration; Instead, it automatically adapts to different configurations. Integrated into the *torch.compile* workflow, the planner observes kernel execution time, communication phases and their duration, available memory headroom, and other behaviors that are determined by the parallelism configurations, and then derives an appropriate DMR plan. Indeed, for all experiments that vary the parallel configuration across models (Figure 11), we use the exact same planner implementation without any per-configuration hand-tuning.
>
> > Q2. How does the checkpointing frequency (n) influence P-DMR coverage and overall verification cost?
> >
>
> When checkpointing frequency decreases from every layer to every n layers, P-DMR coverage reduces to 1/n with proportionally fewer comparisons, without increasing memory usage (Appendix A). To maintain full fault-detection coverage, the remaining (n–1)/n layers are verified through D-DMR or Naive-DMR by design.
>
> We evaluated several AC strategies including per-layer checkpointing and selective submodule (Attention or FFN only) checkpointing, but found them suboptimal. PyTorch's *torch.compile* with *ac_mem_budget* provides automatic, fine-grained operation-level AC that consistently achieves Pareto-optimal throughput-memory trade-offs. We use this approach as our baseline, sweeping multiple memory budgets to select the highest-throughput configuration for each model (Appendix D.2).
>
> As the reviewer correctly notes, there is indeed an interaction between checkpointing frequency and P-DMR coverage. Specifically, the fraction of operations recomputed directly determines the proportion covered by P-DMR. The table below quantifies this relationship for various *ac_mem_budget* values:
>
> | *ac_mem_budget* | 0.0 | 0.25 | 0.5 | 0.75 | 1.0 |
> | --- | --- | --- | --- | --- | --- |
> | Recomputed operations (% of total forward time) | 99.1% | 55.8% | 22.7% | 22.7% | 0.4% |
> | Operations covered by P-DMR (% of total forward time) | 99.1% | 50.9% | 21.3% | 21.3% | 0.4% |
>
> Note that ac_mem_budget (ranging from 0.0 to 1.0) specifies the fraction of activation memory to retain, rather than directly controlling the recomputation ratio. PyTorch's torch.compile selectively checkpoints operations at a fine-grained level to minimize both memory usage and execution time, typically resulting in a lower recomputation ratio than the specified memory budget would suggest.
>
> The table below presents throughput (tokens/s) for Llama-70B with PP=4. Percentages indicate slowdown relative to No-DMR. As expected, lower *ac_mem_budget* values yield higher P-DMR coverage, resulting in greater performance gains over Naive-DMR.
>
> | *ac_mem_budget* | 0.0 | 0.25 | 0.5 | 0.75 | 1.0 |
> | --- | --- | --- | --- | --- | --- |
> | No-DMR | 207 | 226 | 225 | 226 | 229 |
> | naive DMR | 145 (30% ↓) | 158 (30% ↓) | 166 (26% ↓) | 165 (27% ↓) | 169 (26% ↓) |
> | P-DMR | 166 (20% ↓) | 173 (23% ↓) | 176 (22% ↓) | 176 (22% ↓) | 173 (25% ↓) |
>
> > Q3. Is there a quantitative cost model or optimization formulation guiding the planner’s decisions, or are all heuristics empirically tuned?
> >
>
> SpareTrain’s DMR planning combines heuristic components with systematic procedures. We believe this is a practical design choice, as the DMR planning problem reduces to a knapsack-style combinatorial optimization and is therefore NP-hard. Specifically, the high-level prioritization among different strategies (e.g., preferring P-DMR over D-DMR, D-DMR_{inter} over D-DMR_{intra}) is heuristic. However, the application of each strategy is systematic. For example, whether to apply P-DMR is guided by the explicit cost model (Figure 7). Assigning operations to D-DMR_{inter} is done by exhaustively enumerating candidates and selecting the one that yields the largest benefit under time and memory constraints (Figure 9). Finally, assigning operations to D-DMR_{intra} uses a best-effort procedure to cover as many operations as possible while still satisfying the same constraints (Figure 10).

---

> ### Author Response · Authors · 2025-11-21
>
> > Q4. Can the authors provide the distribution of P-DMR, D-DMR, and naïve DMR operators under different configurations to explain the observed throughput changes?
> >
>
> We analyze operation distribution across protection methods and idle window utilization for Llama-70B (batch size=32) under varying memory configurations.
>
> - **Operation Distribution by Protection Method**
>
>
>     | Memory Config | P-DMR | D-DMR | Naive-DMR |
>     | --- | --- | --- | --- |
>     | 80GB (PP=3) | 53.5% | 39.1% | 7.4% |
>     | 94GB (PP=2) | 52.5% | 32.6% | 14.9% |
>     | 141GB (PP=2) | 21.8% | 69.8% | 8.4% |
>
>     80GB and 94GB use *ac_mem_budget* = 0.0, enabling aggressive activation checkpointing that maximizes P-DMR coverage. However, 80GB's PP=3 provides larger PP communication windows, enabling better D-DMR utilization—explaining its superior performance in Section 6.2. 141GB allows higher *ac_mem_budget*, reducing P-DMR coverage. However, abundant memory headroom enables highly effective D-DMR (69.8%), resulting in low end-to-end slowdown.
>
> - **Idle Window Utilization**
>
>
>     | Memory Config | PP Comm Utilization | TP Comm Utilization |
>     | --- | --- | --- |
>     | 80GB (PP=3) | 43% | 53% |
>     | 94GB (PP=2) | 63% | 50% |
>     | 141GB (PP=2) | 74% | 64% |
>
>     **PP Communication:** 80GB's PP=3 configuration generates more frequent PP communication compared to PP=2, resulting in lower utilization percentage despite providing more opportunities for D-DMR. 141GB achieves highest utilization (74%) due to relaxed memory constraints enabling more aggressive AGC segment selection.
>
>     **TP Communication:** Utilization remains relatively consistent. 141GB achieves best efficiency (64%) through optimal Phase 3 fine-grained scheduling enabled by abundant memory.

---

### Official Review · Reviewer_WUDn · 2025-10-31

**Soundness:** 3
**Presentation:** 4
**Contribution:** 3
**Rating:** 6
**Confidence:** 4

**Summary:**

This paper introduces SpareTrain, a system designed to provide complete, low-cost fault tolerance for Large Language Model (LLM) training. The primary goal is to detect Silent Data Corruptions (SDCs) using Dual Modular Redundancy (DMR)—which traditionally incurs a prohibitive 100% compute overhead—at a minimal performance cost. The core contribution of SpareTrain is a set of two complementary strategies that exploit existing characteristics of modern LLM training:

1. Piggyback-DMR (P-DMR): This technique cleverly repurposes the inherent computational redundancy from Activation Checkpointing (AC). The forward pass serves as the primary execution, and the re-computation of activations during the backward pass (which AC necessitates for memory savings) is used as the checker execution.

2. Deferred-DMR (D-DMR): This technique leverages the significant GPU idle time present in large-scale distributed training, which arises from communication overhead (e.g., in Pipeline Parallelism (PP) and Tensor Parallelism (TP)) and pipeline bubbles. SpareTrain defers the checker execution of an operation into these idle periods, effectively hiding its computational latency.

These strategies are orchestrated by a three-phase planner that statically (for dense models) or dynamically (for MoE models) assigns operations to P-DMR, coarse-grained D-DMR (for PP bubbles), or fine-grained D-DMR (for TP/EP communication windows).

**Strengths:**

1. The core strategies are both insightful and effective. The P-DMR concept is particularly creative. Identifying that activation checkpointing already provides a "free" source of redundancy for fault tolerance is a non-trivial and elegant insight. It turns a memory-saving technique into a reliability feature.

2. The paper is exceptionally well-written and easy to follow, especially for a complex systems paper. The concepts of P-DMR and D-DMR are introduced intuitively, and the figures provide clear visual explanations of the baseline, the problem, and the proposed solution's mechanisms.

3. The work is evaluated on relevant, large-scale hardware (H200s) and state-of-the-art models (Llama-3, Mistral-Large, and an MoE model), not just on small-scale toy problems.

**Weaknesses:**

See Questions

**Questions:**

1. Long-term Viability of D-DMR: As distributed training frameworks get better at hiding communication latency and reducing pipeline bubbles, the GPU idle time that D-DMR relies on will decrease. How do you see the effectiveness of SpareTrain changing in a future system that has near-perfect overlap and minimal idle time? Does the benefit of SpareTrain then rely almost entirely on P-DMR?

2. In your experiments on Llama-3 and Mistral-Large (Section 5.1, Figure 7), how frequently did the planner's exception case trigger? That is, what percentage of AC segments were assigned to Naïve-DMR instead of P-DMR because the condition ($2 \times \text{Recompute} < \text{Additional Comm.}$) was met? This would help in understanding the practical importance of P-DMR.

3. Does Silent Data Corruption significantly hurt model quality? Please give us some evidence.

---

> ### Author Response · Authors · 2025-11-21
>
> We appreciate the reviewer's thoughtful comments. We address each point raised in your review below.
>
> > 1. Long-term Viability of D-DMR: As distributed training frameworks get better at hiding communication latency and reducing pipeline bubbles the GPU idle time that D-DMR relies on will decrease. How do you see the effectiveness of SpareTrain changing in a future system that has near-perfect overlap and minimal idle time? Does the benefit of SpareTrain then rely almost entirely on P-DMR?
> >
>
> First, we would like to clarify that SpareTrain utilizes only communication-induced idle time, not pipeline bubbles (Section 3.2). Regarding your question about future systems with reduced communication latency, the short answer is yes: if a future system truly achieves zero GPU idle time, then SpareTrain would rely primarily on P-DMR. However, current systems are far from reaching such an idealized state, and we believe that even future systems will struggle to do so, given that today’s platforms already employ highly aggressive communication–computation overlap techniques. In fact, we have evaluated SpareTrain under asyncTP, the state-of-the-art technique for overlapping communication and computation in distributed training [1] (Appendix E.2). The results remain promising and comparable to those obtained without asyncTP. These results offer insight into how SpareTrain operates under reduced communication latency.
>
> [1] Wang, Shibo, et al. "Overlap communication with dependent computation via decomposition in large deep learning models." Proceedings of the 28th ACM International Conference on Architectural Support for Programming Languages and Operating Systems, Volume 1. 2022.
>
> > 2. In your experiments on Llama-3 and Mistral-Large (Section 5.1, Figure 7), how frequently did the planner's exception case trigger? That is, what percentage of AC segments were assigned to Naïve-DMR instead of P-DMR because the condition was met? This would help in understanding the practical importance of P-DMR.
> >
>
> Here we provide the ratio of exceptions, i.e., the fraction of AC segments that fall back to Naïve-DMR instead of P-DMR. Overall, such fallbacks either do not occur at all or appear only at a low ratio (e.g., under 20%), which highlights the practical importance of P-DMR.
> More specifically, fallbacks are observed only when the memory budget is high (e.g., 141 GB). In such settings, selective rather than full activation checkpointing is applied, resulting in much smaller and more fine-grained AC segments. Since these segments are small, the communication cost is more likely to outweigh the recomputation cost, increasing the chance of triggering a fallback.
>
> | Model | Memory Budget | Exception Ratio |
> |:---:|:---:|:---:|
> | Llama-3-70B | 80GB | 0/80 (0%) |
> | Llama-3-70B | 94GB | 0/80 (0%) |
> | Llama-3-70B | 141GB | 160/880 (18.2%) |
> | Mistral-Large | 80GB | 0/88 (0%) |
> | Mistral-Large | 94GB | 0/88 (0%) |
> | Mistral-Large | 141GB | 176/968 (18.2%) |
>
> We have incorporated the corresponding results and discussion in the revised version (Appendix E.4). Please refer to that section for detailed analysis.
>
> > 3. Does Silent Data Corruption significantly hurt model quality? Please give us some evidence.
> >
>
> Recent large-scale studies demonstrate that SDCs may significantly degrade LLM training outcomes. He et al. [1] demonstrate that even small-magnitude errors can irreversibly degrade model quality, challenging the conventional belief that minor SDCs are harmless. Ma et al. [2] empirically demonstrate that SDC causes silent parameter divergence despite stable training loss, pushing models toward suboptimal minima, even leading to zero test accuracy in extreme cases. Major LLM training reports consistently document SDC-related failures. The Gemini [3], Llama-3 [4], and BLOOM [5] teams all report evidence of SDCs, with some explicitly calling for improved detection mechanisms.
>
> [1] He, Yi, et al. "Understanding and mitigating hardware failures in deep learning training systems." Proceedings of the 50th Annual International Symposium on Computer Architecture. 2023.
>
> [2] Ma, Jeffrey Jian, et al. "Understanding silent data corruption in LLM training." Proceedings of the 63rd Annual Meeting of the Association for Computational Linguistics (Volume 1: Long Papers). 2025.
>
> [3] Team, Gemini, et al. "Gemini: a family of highly capable multimodal models." arXiv:2312.11805 (2023).
>
> [4] Grattafiori, Aaron, et al. "The llama 3 herd of models." arXiv:2407.21783 (2024).
>
> [5] Hugging Face Blog, 2022. https://huggingface.co/blog/bloom-megatron-deepspeed

---

### Official Review · Reviewer_cTfD · 2025-11-02

**Soundness:** 2
**Presentation:** 3
**Contribution:** 3
**Rating:** 4
**Confidence:** 3

**Summary:**

Naive DMR doubles computation to detect silent data corruption. This paper SpareTrain overcomes this through two key innovations: Piggyback-DMR using activation checkpointing, and Deferred-DMR Utilizing idle GPU time and perform checker executions asynchronously. This hides redundancy latency behind existing idle periods, minimizing impact on training throughput. SpareTrain includes both offline and online planners that coordinate P-DMR and D-DMR to balance memory, timing, and redundancy coverage.

**Strengths:**

good originality:
Reusing activation checkpoint recomputation is an creative idea and it works well with existing AC APIs. Deferred-DMR using idle GPU time is also intuitive. The combination of static and dynamic DMR planning introduces a new layer of runtime adaptivity for reliability in distributed training. It's a practical system for achieving full Dual Modular Redundancy (DMR)

good quality:
The paper showcased integration with Llama-3, Mistral-Large, and Llama-4-Scout and up to 32 GPUs. It carefully analyzes overhead sources, memory trade-offs, and timing windows, and conducts ablation studies showing contributions of each design phase. Evaluations demonstrate up to 35% faster than naive DMR with full detection

good Clarity
Checkpointing vs. P-DMR, DMR scheduling diagrams are very helpful to communicate the timing and memory concepts. Despite system-level complexity, the narrative maintains good readability, with consistent terminology and careful explanation of assumptions and constraints.

fair significance
Silent data corruption is becoming a critical reliability issue when scaling to hundreds of thousands of GPUs. Making full DMR feasible at low cost has immediate industrial relevance. SpareTrain’s integration with PyTorch and TorchTitan indicates it could be adopted with minimal workflow disruption

**Weaknesses:**

While the paper’s DMR planner (static + dynamic) is a well-engineered mechanism for allocating operations across Piggyback-DMR and Deferred-DMR, its soundness depends on the assumption that the profiled execution characteristics remain stable throughout training. This assumption is not always valid. The static planner bases its schedule on a short profiling phase that measures activation checkpointing configuration, operation latencies, and communication windows (GPU idle times). However, these metrics can change substantially across iterations due to variations in
* pipeline bubble patterns: which depend on load imbalance, microbatch size, and runtime scheduling
* activation checkpointing overhead: which fluctuates with adaptive memory pressure, gradient accumulation steps, or sequence length changes;

It is less clear what's the net win of full DMR over checkpoint rollbacks. From a cost–benefit standpoint, many production training pipelines might prefer to tolerate occasional checkpoint rollbacks over continuously paying 15% runtime overhead for protection against rare SDCs. Unless the paper provides quantitative evidence that SDCs cause non-recoverable or undetectable model degradation despite checkpointing, the necessity and urgency of system-wide DMR protection remain open to question.

**Questions:**

Would love to see a comparison with checkpoint–resume solution. It remains unclear how frequently such errors materially affect final model quality compared to the dominant sources of variance (e.g., data order, learning rate tuning). When an error or hardware failure occurs, training can resume from the last saved state

---

> ### Author Response · Authors · 2025-11-21
>
> We thank the reviewer for the insightful comments. We respectfully address each concern raised in your review below.
>
> > While the paper’s DMR planner (static + dynamic) is a well-engineered mechanism for allocating operations across Piggyback-DMR and Deferred-DMR, its soundness depends on the assumption that the profiled execution characteristics remain stable throughout training. This assumption is not always valid. The static planner bases its schedule on a short profiling phase that measures activation checkpointing configuration, operation latencies, and communication windows (GPU idle times). However, these metrics can change substantially across iterations due to variations in
> >
> > - pipeline bubble patterns: which depend on load imbalance, microbatch size, and runtime scheduling
> > - activation checkpointing overhead: which fluctuates with adaptive memory pressure, gradient accumulation steps, or sequence length changes;
>
> For a fixed training setup (e.g., (micro)batch size, sequence length, model architecture, and cluster configuration), the training behaviors such as computation time and memory access patterns remain highly stable across iterations. This stability is further pronounced when using graph execution mode (e.g., *torch.compile*), which has become the de facto standard for high-performance LLM training.
>
> Among all components, the only one that exhibits noticeable variation is the duration of inter-GPU communication. Even here, our analysis shows that the fluctuation remains small (typically <5%), which justifies our design choice of using averaged measurements obtained from multiple profiling iterations.
>
> In the rare cases where the training setup changes during execution, such modifications inherently trigger graph reconstruction (i.e., re-invocation of *torch.compile*), and SpareTrain’s planning is consequently re-executed. We find that the overhead of this planning phase is minimal (on the order of a few minutes), which is negligible in realistic scenarios where such re-planning occurs only infrequently.
>
> > It is less clear what's the net win of full DMR over checkpoint rollbacks. From a cost–benefit standpoint, many production training pipelines might prefer to tolerate occasional checkpoint rollbacks over continuously paying 15% runtime overhead for protection against rare SDCs. Unless the paper provides quantitative evidence that SDCs cause non-recoverable or undetectable model degradation despite checkpointing, the necessity and urgency of system-wide DMR protection remain open to question.
> **Questions:** Would love to see a comparison with checkpoint–resume solution. It remains unclear how frequently such errors materially affect final model quality compared to the dominant sources of variance (e.g., data order, learning rate tuning). When an error or hardware failure occurs, training can resume from the last saved state.
> >
>
> First, we would like to clarify that DMR and checkpoint–rollback are not alternative solutions to the same problem; they protect against fundamentally different classes of errors. Computing errors are typically categorized into DCEs (detectable and correctable errors), DUEs (detectable but uncorrectable errors), and SDCs (silent data corruptions, i.e., undetectable errors). Checkpoint–rollback approach handles DUEs. It is inherently incapable of addressing SDCs, because rollback requires knowing that an error has occurred in the first place. In contrast, DMR specifically targets SDCs. Thus, comparing DMR to checkpoint–rollback is not meaningful; the two are orthogonal and complementary.
>
> If you have concerns about whether the (up to) 15% runtime overhead of DMR can be justified, our position is that it is. There is substantial empirical evidence that SDCs do arise in large-scale LLM training and can negatively impact training progress [1,2,3]. The severity of such impact can vary widely depending on where and when the corruption occurs. In benign cases, SDCs may cause only minor perturbations that the inherent robustness of ML training can absorb. But in some cases, they can induce severe model quality degradation. From a practical standpoint, if practitioners are asked whether they are willing to accept a small but non-negligible probability of silently ruining months-long training, versus paying a modest performance overhead (~10%) to eliminate that risk, we believe many would choose the latter.
>
> [1] Bonderson, R. "Training in Turmoil: Silent Data Corruption in Systems at Scale." International Test Conference Silicon Lifecycle Management Workshop, 2021.
>
> [2] He, Yi, et al. "Understanding and mitigating hardware failures in deep learning training systems." Proceedings of the 50th Annual International Symposium on Computer Architecture. 2023.
>
> [3] Dixit, Harish Dattatraya, et al. "Silent data corruptions at scale." arXiv preprint arXiv:2102.11245 (2021).

---

### Meta-Review · Area_Chair_Yfk5 · 2026-01-06

**Summary:**

This paper introduces SpareTrain, a system-level framework designed to detect Silent Data Corruptions (SDCs) in large-scale LLM training at a fraction of the cost of traditional methods. While Dual Modular Redundancy (DMR) typically doubles computational requirements, SpareTrain achieves full protection with only a 3–14% overhead. It accomplishes this by repurposing activation recomputation and exploiting GPU idle time during communication.

In the initial review stage, the reviewers' concerns primarily focused on the practical necessity of DMR compared to simple checkpoint rollbacks, the viability of the system as communication overhead decreases in future hardware, and the analytical depth of the planner's heuristics. To respond to these concerns, the authors provided empirical evidence from Gemini and Llama-3 reports confirming that SDCs are a real threat that rollbacks cannot detect, and they included new ablation data on state-of-the-art asyncTP configurations to prove the system remains effective even with optimized communication. They also provided a quantitative breakdown of operator assignments, showing that P-DMR and D-DMR together cover over 90% of operations across various memory budgets.

Given the solid engineering contribution and the strong performance, this paper is recommended for acceptance.

**Reviewer Concerns:**

The rebuttal effectively addressed the most critical technical concerns regarding real-world necessity and system stability. By providing evidence from Llama-3 and Gemini reports, the authors successfully argued that SDCs are a distinct threat that standard checkpoint-rollbacks cannot detect. They also resolved doubts about overhead and scaling by providing a quantitative breakdown of operator assignments and demonstrating that the system remains effective even with advanced communication-overlap techniques like asyncTP.

Some reviewers did not provide a final follow-up for the rebuttal.

**Reviewer Scores:**

NA

---

### Decision · Program_Chairs · 2026-01-26

Accept (Poster)